# Improving the Management and Treatment of Diabetic Foot Infection: Challenges and Research Opportunities

**DOI:** 10.3390/ijms24043913

**Published:** 2023-02-15

**Authors:** Kaja Turzańska, Oluwafolajimi Adesanya, Ashwene Rajagopal, Mary T. Pryce, Deirdre Fitzgerald Hughes

**Affiliations:** 1Department of Clinical Microbiology, Royal College of Surgeons in Ireland University of Medicine and Health Sciences, Education and Research Centre, Beaumont Hospital, D09 YD60 Dublin, Ireland; 2School of Molecular and Cellular Biology, University of Illinois Urbana-Champaign, Champaign, IL 61801, USA; 3School of Chemical Sciences, Dublin City University, D09 V209 Dublin, Ireland

**Keywords:** photodynamic therapy, antimicrobial peptides, diabetes, DFU, diabetic foot infection

## Abstract

Diabetic foot infection (DFI) management requires complex multidisciplinary care pathways with off-loading, debridement and targeted antibiotic treatment central to positive clinical outcomes. Local administration of topical treatments and advanced wound dressings are often used for more superficial infections, and in combination with systemic antibiotics for more advanced infections. In practice, the choice of such topical approaches, whether alone or as adjuncts, is rarely evidence-based, and there does not appear to be a single market leader. There are several reasons for this, including a lack of clear evidence-based guidelines on their efficacy and a paucity of robust clinical trials. Nonetheless, with a growing number of people living with diabetes, preventing the progression of chronic foot infections to amputation is critical. Topical agents may increasingly play a role, especially as they have potential to limit the use of systemic antibiotics in an environment of increasing antibiotic resistance. While a number of advanced dressings are currently marketed for DFI, here we review the literature describing promising future-focused approaches for topical treatment of DFI that may overcome some of the current hurdles. Specifically, we focus on antibiotic-impregnated biomaterials, novel antimicrobial peptides and photodynamic therapy.

## 1. Introduction

The International Diabetes Federation estimates that 537 million people were living with diabetes in 2021, representing 10% of the global population, which has increased by 1.2% in the past five years. It is predicted that this number will rise to 783 million by 2045 [1]. The development of diabetic foot ulcers (DFUs) is a complication of diabetes that is estimated to affect 9.1–26.1 million people annually with a global prevalence of 6.3% [2]. One Malaysian study reported an infection rate of DFUs of 41.5% [3], with a similar value of 40.1% being reported by an Australian study [4]. Wound infection in those with DFUs is among the strongest predictors of lower limb amputation [5], and more than one million diabetic patients suffer lower limb loss annually, as a result of failure of therapeutic interventions for treating diabetic foot infections [6]. Notably, the three-year mortality rate after the first amputation is between 20% and 50% [7]. In addition to obvious physical morbidity and mortality, DFUs also present significant psychological challenges for patients, including depression, emotional distress and anxiety disorders [8,9,10].

Diabetic foot infection (DFI) is the commonest cause of hospitalisation among those with diabetes [11] and the complexity of management may result in lengthened hospital stays and subsequently increased healthcare costs. Therefore, to ensure better clinical outcomes for patients, there is a critical need to effectively manage and treat these infections. In this review, we explore the unique challenges of resolving DFIs and outline the current clinical management approaches in relation to wound healing and antimicrobial therapy. We also describe alternative novel strategies that are currently under development and explore their potential to overcome the therapeutic challenges of DFI and improve outcomes for patients.

## 2. Host Factor and Infections of Diabetic Foot Ulcers

Diabetes-associated peripheral neuropathy impairs the distal nerves of the limbs, resulting in reduced pain sensations and progressive numbness. As a consequence, diabetic foot ulcers may develop and progress unrecognised following relatively minor external traumas. Irregular plantar pressure and shear stress on the foot associated with reduced foot mobility and abnormal structural features such as hammer toe and Charcot deformity may further facilitate ulcer progression [12]. Once a DFU becomes infected, the most important factor determining clinical outcome is peripheral arterial disease (PAD), a manifestation of atherosclerotic disease defined as the partial or complete obstruction of one or more arteries [13]. The rate of PAD is 2–4-fold higher in patients with diabetes [14]. PAD-associated vascular complications in diabetes contribute to perfusion deficits that reduce blood flow to infected tissue, resulting in significantly poor resolution of infected ulcers by limiting the supply of immune cells and mediators to DFIs. In addition, the innate immune responses that normally protect from infection are compromised by acute hyperglycaemia in diabetes. Several defects in neutrophil function in diabetic animal models and human studies have been shown, including reduced respiratory burst [15], limited phagocytic activity [16] and downregulation of toll-like receptors for pathogen recognition [17].

## 3. Complications and Risks for Poor Clinical Outcomes

The symptoms of wound infection commonly include purulent exudate, odour, erythema, warmth, tenderness, oedema, pain, fever and elevated white cell count. However, these clinical signs of infection may not all be present in the immunocompromised patient, the patient with poor perfusion or the patient with a chronic wound. The latter three conditions are common among diabetic patients, making DFIs inherently complex to diagnose, categorise and manage—requiring a multidisciplinary care approach involving podiatrists, vascular surgeons, tissue viability experts and clinical microbiologists. Furthermore, multiple classification schemes for diabetic foot complications exist, of which infection is often a defined parameter in the broader classification of ulceration severity. However, because infection severity is a major contributor to poor clinical outcomes, the classification system recommended by the Infectious Diseases Society of America (IDSA) and the International Working Group of the Diabetic Foot (IWGDF) shows utility in a clinical setting as a predictor of lower limb amputation [18,19]. This system is a modification of the PEDIS classification system, which assesses five main wound parameters: perfusion (P), extent (E), depth (D), infection (I) and sensation (S) [20,21].

The emerging trend towards greater prominence of PAD among people with diabetes adds further complexity to the pathophysiology of foot wounds, meaning that such wound characteristics as contained within the PEDIS classification may all contribute to patient clinical outcomes and ought to be assessed while classifying DFIs. The Society for Vascular Surgery’s integrated system of classification of the lower extremity threatened limb is based on Wound, Ischaemia and Foot Infection (WIfI) to better predict amputation risk [22]. Recently updated guidelines continue to recommend these two classification systems to guide infection management and support clinical decision making [23].

## 4. Aetiology of Diabetic Foot Infections

In the clinical management of DFUs, it is important to differentiate between colonisation and infection. Colonisation is defined as the presence of multiplying bacteria without an overt host immune response and is a feature of chronic wounds, usually involving bacteria of the normal flora [24]. Approximately one-half of DFUs become infected [25]. Therefore, understanding the complexity and microevolution of microbial aetiology in the environment of the diabetic foot is important in infection diagnosis and management. The transition from colonisation to infection occurs when colonising microorganisms invade the deeper tissues and overwhelm the protective immune defences. In the setting of diabetes, dysfunctional neutrophils and morphological changes in macrophages secondary to hyperglycaemia contribute to infection risk. According to the most recent recommendation from the IDSA, the diagnosis of clinical infection (a pre-requisite for antibiotic therapy), as distinct from wound contamination or colonisation, requires the presence of at least two signs or symptoms of inflammation and/or the presence of purulent discharge [26].

Although the aetiology of DFUs and DFIs are complex and often polymicrobial, Gram-positive cocci, mainly *Staphylococcus aureus*, remain the most common bacterial pathogen isolated from DFIs, followed by aerobic Gram-negatives, mainly *Pseudomonas aeruginosa* [27,28]. Other isolated bacteria include Enterococcus Spp., Enterobacterales and Group B *Streptococci*. For lower extremity infections, bacterial burden of ≥10^4^ colony-forming units (CFU) per gram is considered the critical load at which infection becomes inevitable [29,30]. More recently, for DFI, lower thresholds have been shown to be clinically significant for specific organisms such as toxin-producing β-haemolytic Streptocooci or *P. aeruginosa* [31,32]. Xu et al. showed a delay in wound healing of up to 44% in diabetic neuropathic ulcers for each log increase in microbial CFU, as well as an association between poor wound healing and a 10^4^ CFU threshold [33]. Primary infection of DFUs and shorter-duration infections generally involve one type of organism, but long-standing chronic infections, which are common in diabetic patients, tend to be polymicrobial, including both aerobic and anaerobic bacteria species. In addition to the multi-species (and multi-genera) characteristics of more advanced DFIs, the continuum of infection progression inevitably involves the production of biofilm [34]. Biofilm production facilitates the microbial aggregation, bound by a polysaccharide-, protein- or nucleic acid-rich extracellular matrix. Bacteria in biofilms are protected from the host immune responses and the effects of antibiotics based on the biofilm’s physical characteristics, the slower metabolism of cells within the structure and complex cell-to-cell communication [35,36]. Hence, many DFIs prove recalcitrant to conventional antibiotic therapy.

Given the complexity of both host and pathogen factors that may contribute to poor clinical outcomes in DFI, a multidisciplinary approach is the cornerstone of management with early intervention key to reducing risk of severe infection and amputation. This approach can include, for example, foot care and regular foot inspection for ulcer prevention and off-loading devices for treatment of ulcers caused by biomechanical stress. Debridement, the removal of slough and necrotic tissue, can promote healing and limit infection severity [37]. The choice of antibiotics for DFI is also complex, and factors including severity, causative pathogen/s, antibiotic resistance, oral bioavailability and achievable bone concentration must be considered [38]. Risk-reducing management and treatment require considerable and regular patient engagement with a range of medical, surgical and other healthcare services. Some treatments, e.g., surgery and systemic antibiotics, require patients to be hospitalised. Increasing levels of antimicrobial resistance (AMR) among DFI pathogens can also result in treatment failures. Consideration of novel and alternative approaches that mitigate the risk of severe infection or that facilitate self-management where possible should be considered.

## 5. Topical Antimicrobials for Treating DFIs

The 2012 guidelines of the IDSA recommended debridement and systemic antibiotics as the standard treatment for clinically infected wounds [26]. Topical treatments are recommended as monotherapy for more superficial infections, or in combination with systemic antibiotics for more advanced infections [39]. However, based on the continued lack of published studies to support their efficacy for treating DFIs, in the 2019 update to the 2016 IWDGF Guidelines on Management and Treatment of Foot Infection in Persons with Diabetes [40], current topical agents are not recommended as adjuvants for DFI treatment [23]. In theory, topical antibiotics for DFI have the potential to deliver high antimicrobial concentrations at the site of infection, bypassing disease-associated vascular insufficiencies. Several systematic reviews have been undertaken in this area with variable conclusions regarding the efficacies of systemic vs. topical modalities [39,40,41,42,43]. A systematic review of antimicrobial agents for chronic wounds, including DFUs, concluded that few systemic agents improved outcomes, but faster healing rates were achieved with the use of several topical substances [41]. Notably, Peters et al. reported that a novel antimicrobial peptide, Pexiganan, in a cream formulation, was similar in effectiveness to a systemic antibiotic (ofloxacin) in the treatment of mildly infected diabetic foot [44]. Despite the current paucity of randomised controlled trials of topical agents for DFIs, they potentially offer many benefits in this context, including local delivery of high and sustainable antimicrobial drug concentrations, reduced systemic absorption and lowered risk of toxicities [45]. More convenient application in an out-patient setting may also contribute to greater adherence rates and better treatment outcomes [46]. Indeed, a large retrospective cohort study involving 1378 patients showed more than 200% improvement in healing at any time point in those receiving personalised topical treatment guided by molecular diagnosis compared to those receiving standard-of-care treatment [47].

## 6. Strategies under Investigation with Potential in DFI Management

### 6.1. Antibiotic Impregnated Biomaterials

Adjunctive antimicrobial wound dressings are available clinically for inclusion in the management of DFIs and other chronic wound infections, but the choice of dressing is often patient or clinician experience-driven rather than evidence-based. Most of these dressings incorporate either iodine, silver or chlorhexidine, but quality randomised controlled trials are lacking to establish a clear evidence base. The properties of the dressing materials also require critical consideration of the nature of the infected wound and may differ for sloughy compared to granulating wounds. Dressing materials in current use include tulle, alginate, polyurethane foams, hydrocolloids and hydrogels [46,47,48].

In advancing topical treatment with antimicrobials, e.g., silver sulfadiazine (SSD) or conventional antibiotics, such as tetracycline or gentamicin, a wide range of functionalised biomaterials have been developed as targeted delivery vehicles for localised antimicrobial activity. Many developed in the last decade have been designed with additional desirable features such as antioxidant, anti-infective, debridement, immunoregulatory, and angiogenic properties [43,44,45,46,47,48,49,50,51]. The potential efficacy of topical antibiotics in the treatment of DFIs or chronic wound infections has been explored in preclinical and clinical studies, with a few already being used clinically for patient care. For example, the silver sulfadiazine (SSD)-loaded chitosan microspheres (CSM) developed by Seetharaman et al. [52] in 2011 use a water-in-oil emulsion technique to impregnate SSD-CSM particles with polyethylene glycol fibrin gels. These gels showed in vitro bacterial killing with minimum inhibitory concentration (MIC) of 125 μg mL^−1^ and 100 μg mL^−1^ against *S. aureus* and *P. aeruginosa*, respectively, and have been translated to clinical use with encouraging results [53]. More recently, Pérez-Díaz and colleagues showed that chitosan gel formulations loaded with silver sulfadiazine were effective antimicrobial agents against methicillin-resistant *S. aureus* (MRSA) and *P. aeruginosa* within formed biofilms obtained from patients with chronic wound infections [54].

Calcium sulphate balls are a class of biodegradable carriers explored as functional antibiotic-impregnated biomaterials for treating DFIs and chronic wound infections. They have an excellent elution profile and can be impregnated with a broad range of water-soluble antibiotics including aminoglycosides, vancomycin, fluoroquinolones and daptomycin [45]. In a case study, tobramycin-impregnated calcium sulphate pellets were inserted into the diabetic foot ulcer cavity and used in combination with oral antibiotics for the successful treatment of forefoot osteomyelitis in a patient with diabetes [55]. Both Karr [56] and Morley et al. [57] have reported similarly successful results with vancomycin-loaded calcium sulphate and hydroxyapatite beads in the treatment of diabetic forefoot osteomyelitis. A significant benefit in the topical administration of antibiotics was shown by Price et al., who compared the in vitro efficacy of oral antibiotics with that of gentamicin-loaded calcium sulphate beads against *P. aeruginosa*, *S. aureus* and a polymicrobial biofilm containing bacteria harvested from diabetic foot infections [58]. They showed a 5–9 log reduction in microbial burden with the topically administered gentamicin-loaded calcium sulphate compared to only 0–2 log reduction for systemic antibiotic administration [58]. Biodegradable materials such as calcium sulphate offer significant clinical advantages by eliminating the need for surgical removal. Superior elution profiles have also been reported compared with the earlier non-biodegradable polymethylmethacrylate (PMMA) beads, which had poor sustained antibiotic release after the first 48–72 h of application [59,60].

While local delivery of antibiotics using these functional biomaterials offers a promising alternative and/or adjunct to systemic antibiotics for treating DFIs, well-designed clinical studies would further establish their efficacy under varying clinical conditions. While less complex regulatory pathways may exist for antibiotics with well-known safety and efficacy profiles than for biomaterials based on entirely new agents, the rates of antibiotic resistance among bacteria that cause DFI are likely to increase, presenting a further consideration in the development of antibiotic-based topical solutions.

### 6.2. Antimicrobial Peptides and Peptide-Based Polymers

Antimicrobial peptides (AMPs) are endogenous molecules that function in the innate immune system of different organisms, where they serve as a natural first line of defence against infections caused by a broad range of microbes, including Gram-positive and Gram-negative bacteria, anaerobic organisms, viruses and fungi [61]. Most AMPs are cationic in nature, initiating electrostatic interactions with their negatively charged phospholipid in the cell envelope of their bacterial targets (cytoplasmic membrane and Gram-negative outer membrane) [62]. AMP-mediated disruptive effects on the cell membrane vary and include the formation of membrane-spanning pores, distorted membrane curvature, the formation of aggregates or membrane dispersal, all of which are well described elsewhere [63].

In addition to their antimicrobial properties, natural AMPs also play important roles in wound healing. The AMPs LL-37 and β-defensins have been studied for wound healing activity and their effects include stimulating cytokine/chemokine production, promoting keratinocyte migration, proliferation and angiogenesis [64,65]. Chitosan hydrogel encapsulation of LL-37 stimulated effective wound repair with improved biocompatibility in a mouse wound model [66]. Other naturally occurring peptides, particularly those from various frog species, including Temporin-A (*Rana temporaria*) and Citropin 1.1. (*Litoria citropa*), show potent antimicrobial activity in vitro against 30 *S. aureus* (range 1–4 and 1–8 mg/L) and 30 *P. aeruginosa* isolates (range 8–64 and 2–16 mg/L) recovered from wounds [67]. A small 11-amino-acid peptide, CW49, promoted healing of a full-thickness skin wound in diabetic animals by mediating the upregulation of angiogenesis [68]. Anti-biofilm properties are critical in managing DFI, and AMPs have shown promising results in this regard. Another derivative of the frog skin AMP, esculentin-1, Esc (1–21)-1c, showed biofilm eradication concentrations of 12.5 μM against *P. aeruginosa* strains, comparable to that of colistin in vitro [69].

Investigation of AMPs for wound applications remains focussed largely on pre-clinical studies, the exception of which is pexiganan, the magainin derivative which progressed as a 1% cream formulation into phase III clinical trials, although it failed to meet the primary clinical endpoint of superiority over placebo [70,71] and had no clear advantage over systemic antibiotics. It is likely that harnessing the anti-polymicrobial and wound-healing properties of AMPs for applications in wound management will require chemical modifications and/or the use of novel delivery systems to reduce cytotoxicity and increase stability or biocompatibility while improving AMP targeting and prolonging their activity at the wound site. Collagen functionalised with LL-37 or its derivatives maintained antimicrobial activity and low cytotoxicity in vitro [72]. Notably, in terms of targeted delivery, Casciaro et al. engineered a derivative of the frog skin AMP, esculentin-1, Esc (1–21), for topical applications in wound infection. When conjugated to gold nanoparticles using polyethylene glycol as a linker, AuNPs@Esc (1–21) demonstrated wound-healing in a keratinocyte model and up to a 15-fold increase in anti-biofilm activity against *P. aeruginosa* with low cytotoxicity [69,73]. The amino acid homopolymer, ε-poly-L-lysine (ε-PL), composed of L-lysine residues, has shown broad-spectrum antimicrobial activity with wide-ranging potential applications in health and medicine [74]. Its application for wound infection management has been explored by integrating εPL into nanofiber dressings. In this form, it accelerated wound healing of bacteria-infected burns compared to standard-of-care silver dressings, in addition to low propensity for antimicrobial resistance development and high biocompatibility for skin cells [75].

Some of the limitations of natural AMPs, such as high production costs, stability issues and biodegradability, have been addressed by developments in synthetic antimicrobial polymers and peptide analogues with structures inspired by AMPs. These comprise polymer backbones with key antimicrobial structural features such as cationic charge and hydrophobic groups. One promising group is structurally nanoengineered antimicrobial peptide-based polymers (SNAPPs). They are star-shaped polypeptide nanoparticles usually rich in positively charged amino acids such as lysine, valine or arginine residues with multimodal bactericidal mechanisms involving lipopolysaccharide (LPS) targeting and outer membrane destabilisation or fragmentation [76]. Similar to linear AMPs, this results in lethal disruption of membrane function, along with membrane depolarisation or hyperpolarisation, and oxidative stress through the production of reactive oxygen species (ROS), all of which ultimately result in microbial cell death [77] (Figure 1). Shirbin and colleagues reported that architectural modifications influenced the antimicrobial activity of star-shaped SNAPPs [78]. Through random opening polymerisation (ROP), they designed SNAPPs of varying arm numbers, and reported that increasing either the arm length or arm number resulted in greater antimicrobial activity [78]. They attributed this result to increased local concentration of polypeptide arms, but also observed that SNAPP cytotoxicity increased with increasing arm length and arm number. Through therapeutic index calculations, they identified the S16 and S4 structural isoforms to be the best therapeutic agents with the lowest toxicities [78]. Recently, our group demonstrated, using poly-L-lysine (PLL)-based SNAPPs, no statistically significant difference in bactericidal or anti-biofilm activity against *S. aureus* and *P. aeruginosa* reference strains for linear *Vs* star arrangements. Furthermore, using the linear form, PLL160, we showed 3.04 ± 0.16 and 3.96 ± 0.83 log reduction in CFU/mL of *S. aureus* and *P. aeruginosa* recovered from diabetic foot infections [79]. Unlike most antimicrobial therapeutics which only show direct bacterial killing, several studies have also shown that SNAPPs also possess potent immunomodulatory properties, stimulating the body’s innate immune response by recruiting neutrophils and monocytes to neutralise bacterial toxins and promote effective wound healing and angiogenesis [80,81]. These features make SNAPPs attractive candidate molecules for development as therapeutics for DFUs complicated by chronic antibiotic resistant infection.

### 6.3. Synergistic Antimicrobial Activity of Antimicrobial Peptides and Polymers with Antibiotics

One application of AMPs and synthetic AMP-based polymers that may have benefits in DFI management is their use as adjuncts to antibiotic therapy for maximal therapeutic efficacy. Such synergistic approaches would be critical in preventing the emergence and spread of bacterial resistance mechanisms against either entity, while re-sensitising previously resistant bacteria to antibiotics agents in an evolutionary trade-off. Using a checkerboard assay, de Gier and colleagues mapped the synergistic action of BA250-C10, a synthetic lipidated antimicrobial peptide polymer with either colistin or tobramycin, and showed that the combination effectively inhibited the in vitro growth of *P. aeruginosa* strains obtained from patients with cystic fibrosis, to a much greater extent than either agent could achieve alone [82]. Similar results were obtained by Amani et al., who showed that the antimicrobial peptide CM11 could be used in synergy with antibiotics against a range of clinically relevant bacteria strains, including *S. aureus, P. aeruginosa, Klebsiella pneumoniae* and *Acinetobacter baumannii* [83]. C12(ω7)K-β12, another lipopeptide-like synthetic molecule, could be used to rapidly induce a transient state of membrane depolarisation, which disrupted the proton-motive force required by bacterial antibiotic efflux pump and re-sensitised these previously resistant bacteria strains to intracellularly acting antibiotics [84].

In summary, AMPs and AMP-based polymers are effective candidate antimicrobial molecules against clinically relevant antibiotic-resistant bacteria, which complicate most DFUs. Their multimodal mechanism of action may limit the emergence of sophisticated resistance mechanisms in bacteria compared to conventional antibiotics. In addition, they have potential to re-sensitise previously antibiotic-resistant bacterial strains by disrupting bacterial membrane potential. AMP or AMP-based polymer-mediated immunomodulatory properties can promote wound healing and angiogenesis, and their unique structural properties allow for the fine-tuning of their antimicrobial action, making them promising alternatives and/or adjuncts to currently available DFUs/DFIs treatment options.

### 6.4. Photodynamic Therapy

Photodynamic therapy (PDT) employs a light source to generate reactive oxygen species via an excited photosensitiser (PS). This approach has been applied successfully in multiple clinical contexts. Photofrin^R^, a porphyrin-based material, was the first FDA-approved PS for treating a number of cancers, including oesophageal and skin cancers [85]. PDT is also used in chronic recurrent sinusitis, periodontitis, and for a range of other ophthalmological and dermatological indications [86,87]. Many compounds have been approved clinically as effective photosensitisers. In oncology, these are mainly tetrapyrrole compounds such as porphyrins, chlorins, bacteriochlorins and phthalocyanines. Other PDT compounds include phenothiazinium salts, methylene blue and toluidine blue O, which have been used in clinical trials for cancer chemotherapy and have been reviewed elsewhere [88].

There is growing interest in the antimicrobial applications of PDT for the topical treatment of infected wounds [88,89,90]. Antimicrobial PDT involves administering a non-toxic PS that is selectively accumulated by bacteria and not the surrounding tissue, followed by photo-activation with light of an appropriate wavelength to generate ROS such as free radicals and singlet oxygen, which are cytotoxic to bacterial cells [91]. Generation of ROS fromPSs is an oxygen-dependent process that takes place via a type I or type II reaction pathway. In the type I pathway, photo-induced interactions between the PS and biological substrates, such as amino acids and nucleic acids, produce superoxides by hydrogen or electron transfer. The resulting oxidative stress, particularly to lipid-rich bacterial cell membranes, causes irreversible damage and kills the bacteria cell. The type II pathway generates predominantly singlet oxygen by triplet energy transfer reactions between the PS and molecular oxygen [92]. As PSs can potentially bind to many sites in the bacteria cell, the lethal oxidative damage caused by PDT is non-specific. This is important for activity against AMR pathogens and greatly reduces the likelihood of developing resistance, as confirmed by several laboratory experiments [93,94,95]. In addition, anionic or neutral PS also damage DNA strands, lowering thymidine incorporation during DNA synthesis [96].

The favourable results from the in vitro application of antimicrobial PDT are well described, but fewer data are available for its in vivo preclinical application in animal models or clinical studies. Tanaka et al. compared the in vitro bactericidal activity of a commercial PS, Photofrin1^TM^, to its activity in a murine MRSA arthritis model. They found that in vivo, too low or too high light doses were ineffective, and they noted the destruction of neutrophils at higher light doses [97]. Further work from these authors showed that the effects on murine neutrophils at higher light doses was PDT-type dependent with the more hydrophilic PDTs, e.g., toluene blue- or methylene blue-based PDTs, having the least toxicity to neutrophils [97]. Work by Zolfaghari et al. using a mouse wound model demonstrated <2 log reduction in CFU/mL of *S. aureus* in wounds following application of methylene blue-based PDT [98], whereas others have reported much higher efficacy levels of up to 5 log reduction using the same PDT in vitro, under comparable conditions of activation [99]. Similar differences in in vivo and in vitro antimicrobial efficacies have been reported for other PDTs [100]. Some examples of novelPSs that have shown relatively high in vitro activity that translated to promising in vivo activity in wound models of infection are summarised in Table 1.

A phenothiazinium derivative, PPA904 [3,7-bis(N,N-dibutylamino) phenothiazin-5-ium bromide], progressed to a phase IIb controlled trial for applications in the care of infected wounds, including DFI. While only a small study of eight patients, a statistically significant reduction in bacterial load (approximately 1 log immediately after PDT treatment) and a trend towards wound healing were observed. The authors reported that the treatment was well-tolerated with an obvious improvement in wound healing within the treatment group compared to the placebo group [94,101]. Importantly, the authors noted the convenience and ease of use of PDT in the clinical environment, as reported by the healthcare workers involved in the trial. One Brazilian trial on a small number of patients (*n* = 18) showed a reduction in the frequency of amputations in patients treated with PDT using phenothiazinium salts compared to the control group, reflecting increased rates of healing of the DFUs [102].

Another group of molecules that offer exciting prospects for the application of antimicrobial PDT for wound applications are those with the 4,4-difluoro-4-bora-3a,4a-diaza-s-indance structure, commonly referred to as BODIPYs. They possess desirable photophysical properties including large molar extinction coefficients in the visible light region, long emission wavelength and photostability. In addition, their chemical versatility facilitates the structural modification of these molecules to increase their production of singlet oxygen species. This is achieved by introducing heavy atoms such as iodine or bromine into their structure to create a high-efficiency triplet excited state, which ensures the efficient production of singlet oxygen [103]. This approach is, however, associated with its own challenges, including the reliance on halogens for antimicrobial activity and dark activity [104]. Li et al., recently reported the designs of a multifunctional BODIPYPS containing a positively charged guanidine group, LIBDP, with bactericidal properties that caused membrane destruction and inhibition of microbial proliferation of Gram-positive bacteria [105]. This PS generates reactive oxygen species upon light activation that destroyed pre-formed biofilms and oxidised nitric oxide in wounds to promote wound healing [105].

Lin et al. investigated structure–activity relationships of BODIPYs containing *meso*-Pyridinyl and methyl pyridinium or *meso*-pyridinium BODIPYs carrying hydrophobic head groups against *S. aureus*, *Escherichia coli*, *Candida albicans* and MRSA. Methyl-meso-(meta-pyridinium) BODIPY showed the highest phototoxicity against these pathogens with MICs ranging from 0.63 to 1.25 μM at a light dose of 81 J/cm^2^. This BODIPY group showed 3.59 to 7.05 log reduction in CFU/mL of *S. aureus*, including MRSA at concentrations from 0.31 to 5 µM, and 2.53 to 6.62 log reduction in CFU/mL of *E. coli* at concentrations from 1.25 μM to 5 μM. An in vivo aPDT study performed in a mouse *S. aureus* wound model using the methyl-meso-(meta-pyridinium) BODIPY formulated into a hydrogel to treat the infected area showed visible wound healing and significant reduction in bacterial load within 8 days.

Reducing the bacterial burden of chronic ulcers helps to reduce the risk of infection but also removes a potential barrier to effective wound healing. Conventional antibiotics have limited efficacy due to poor tissue penetration, increasing bacterial resistance and the risk of allergic contact reactions with topical antibiotics. In contrast, aPDT can promote healing by modulating growth factors production even in the absence of any antimicrobial activity. Statius van Eps et al. showed that PDT in a vascular smooth muscle cell injury model, stimulated endothelial cell growth by inactivating transforming growth factor-beta (TGF-β). However, the authors cautioned that further work is required to establish optimal dosing, noting inhibition of mediators of wound healing with sub-therapeutic PDT [106]. More recently, Mills and colleagues showed that topical application of methyl aminolevulinic acid (MAL)-photodynamic therapy specifically increased TGF-β3 and metalloproteinase 1 and 9 production to stimulate matrix remodelling and the production of scars with improved dermal matrix architecture following excision skin wounding [107].

**Table 1 ijms-24-03913-t001:** Examples of novel photosensitisers with antimicrobial PDT activity in vitro and in vivo.

Chemical Structure/Name	Antimicrobial Activity ^a^	Irradiation Wavelength ^b^	Singlet Oxygen (Φ_Δ_) ^c^	Reference
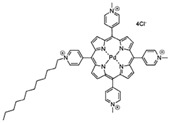 Meso-substituted tetra methyl-pyridinium porphyrin	In vitro ≥7 log killing of *S. aureus* (10 nM), *E. coli* (100 nM), *C. albicans* (500 nM) In vivo Mouse wound infection model 4 log killing of *E. coli* Recurrence prevented up to 5 days post-aPDT	405 nm	0.71	[108]
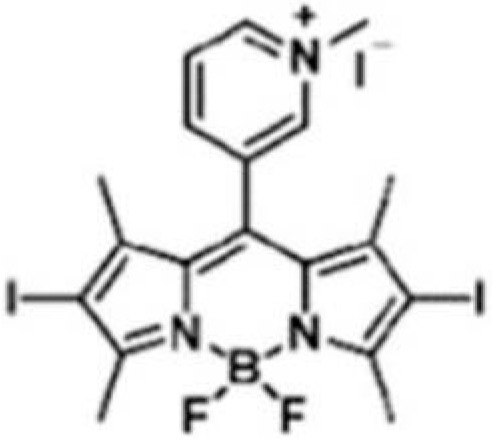 Methyl meso-(meta-pyridinium) BODIPY	In vitro 7 log killing of *S. aureus* (5 μM, 15 min) 6 *E. coli* (5 μM, 15 min) 3 log killing of *C. albicans* (1 μM, 15 min) In vivo Mouse wound infection model. Increased wound healing and significantly reduced viable *S. aureus* by day 5.	>450 nm	0.84	[109]
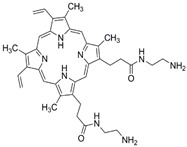 N, N′-bis (2-aminoethyl)-2,7,12,18-tetramethyl-3,8-divinyl-21H, 23H-porphyrin13,17-bispropanamide porphyrin	In vitro 4 log killing of *P. aeruginosa* (50 μM, 30 min) In vivo Mouse wound infection model. 4.2 log killing of *P. aeruginosa* (100 mM) post-aPDT. Statistically significantly increased wound healing at day 10.	650 nm	0.68	[110]
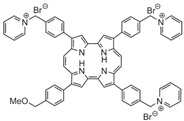 2,7,12-Tris(r-pyridinio-p-tolyl)-17-(p-(methoxymethyl)phenyl) porphycene	In vitro 6 log killing of *E. faecalis* (0.5 μM) 6 log killing *S. aureus* (2 μM) 6 log killing of *A. baumanni* (8 μM) 6 log killing of *E. coli* (10 μM) In vivo Mouse burn infection model 2.6 log reduction in MRSA (100 μM)	652 nm	0.03	[111]
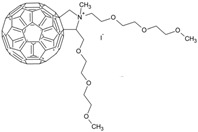 Mono-N-methylpyrrolidinium fullerene iodide	In vitro >6 log killing of *S. aureus* (10 μM) 3 log killing of *E. coli* (100 μM) 1 log killing of *P. aeruginosa* (100 μM) In vivo Mouse infection model. Reduction in *S. aureus* viability 3 days post-aPDT. Moderate wound healing effects.	385–780 nm	0.31	[112]

^a^ A summary of the main activities is given, along with the microorganisms investigated, their level of killing (log reduction in colony-forming units (CFU)/mL) and the concentration of compound tested. For in vivo testing, summary is of the animal model, bacteria investigated and main findings. ^b^ The wavelength of light used for compound activation. ^c^ Reported quantum yield of singlet oxygen, where quantum yield is the number of molecules undergoing a photochemical event per absorbed photon.

### 6.5. Combination of PDT with Antibiotics

Interestingly, a synergy between PDT and penicillin antibiotics has been demonstrated in some studies, suggesting its potential as an adjunct to antibiotics. Iluz et al., using a checkerboard method, demonstrated synergistic antimicrobial activity against *S. aureus* using deuteroporphyrin (DP)-PDT with oxacillin but not with vancomycin, gentamicin, fusidic acid or rifampicin [113]. Additionally, they reported that while DP-PDT alone resulted in a 3–6 log reduction in viability of cells within mature *S. aureus* biofilms, its efficacy further increased by 3 log reduction when combined with oxacillin. The authors speculated that the impaired penicillin resistance among PDT survivors may reflect damage to plasmids that mediate beta-lactamase production based on earlier studies showing damage to supercoiled plasmid DNA in response to PDT [96].

Photodynamic therapy has the potential to either reduce infection burden or eradicate infection completely. Its multiple bacterial targets may both abrogate existing resistance of pathogens to conventional antibiotics while limiting the development of new resistance mechanisms. The flexibility of feasible structural modifications within PS structures are also attractive for the future development of technologies towards wound healing applications for DFUs and other infected wound types. With well-designed clinical and preclinical studies, photodynamic therapy could be progressed into clinical practice for the treatment of multi-drug resistant wound infections. Perhaps the most appealing aspect of antimicrobial PDT is its potential for use in outpatient clinics at relatively low costs. Hospitalisation may not be required when using PDT, which reduces the cost of treatment by several orders of magnitude. It is also associated with the use of already available light sources and inexpensive photosensitisers.

## 7. Conclusions

The management of infected DFUs is a critical component of the holistic management of the diabetic patient. Numerous approaches including antimicrobial peptides and peptide-based polymers, photodynamic therapy and antibiotics-impregnated biomaterials have been investigated in preclinical studies, with promising antimicrobial activity often against bacteria in biofilm growth modes. However, there is an urgent need to progress them into the clinic, through well-designed randomised controlled trials to generate much-needed evidence of clinical efficacy. These approaches present multimodal benefits of being applicable both as stand-alone therapies and as adjuncts to long-standing antibiotics, which have become less potent in the wake of rising antibiotic resistance. They also offer the advantage of being effective both as antimicrobial agents and promoters of wound healing—a benefit that cannot be overstated in the setting of diabetic foot infections.

## Figures and Tables

**Figure 1 ijms-24-03913-f001:**
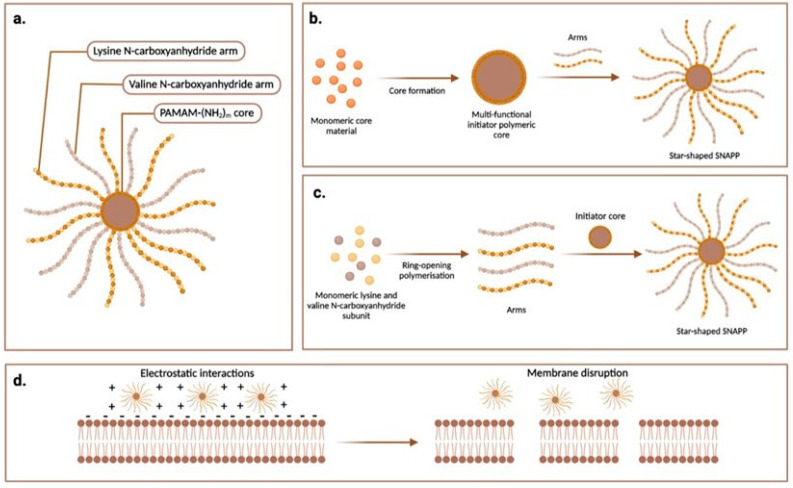
Design features and novel bacterial membrane-specific interactions of structurally nano-engineered antimicrobial polypeptide polymers (SNAPPs): (**a**) Example of a SNAPP co-polymer structure, comprising a multifunctional initiator core of poly(amidoamine) or PAMAM, with *m* number of lysine or valine N-carboxyanhydride (NCA) polymeric arms complexed to its amino (NH_2_) terminus. *m* could be 16 or 32 (**b**) ‘*core-first approach*’ for SNAPP synthesis, involving synthesis of PAMAM core from monomeric amidoamine starting materials followed by the complexing of polymeric lysine or valine NCA arms. (**c**) ‘*Arm-first approach*’ for SNAPP synthesis, involving initial synthesis of the arms by ring-opening polymerisation of monomeric units of lysine or valine NCA. (**d**) Antimicrobial action of SNAPPs involves electrostatic interactions between their negatively charged arms and the positively charged bacterial cytoplasmic membrane, leading to membrane disruption, penetration, pore formation and cell lysis. Figure constructed using the web tool at www.biorender.com.

## Data Availability

Not applicable.

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
