# Peer review of "Improving the Management and Treatment of Diabetic Foot Infection: Challenges and Research Opportunities"

_ijms, 2023, doi:10.3390/ijms24043913_

Round 1

Reviewer 1 Report

Brief Summary

The article by Turzanska et al offered a nice review of the current landscape and clinical management approaches of diabetic foot infection (DFI). The article reads clear and well organized. The review discussed and compared a few important current and promising novel treatment options of DFI, which calls for further investigation into these very promising categories of new treatment options, including antibiotic impregnated biomaterials, antimicrobial peptides and peptide-based polymers, as well as photodynamic therapy. These new treatment options have the potential to be safer, more efficacious, easier to perform, less likely to cause drug resistance and supporting better wound healing. The article provides a nice reference for future researchers to advance the field.

General Concept Comments

1.     The introduction nicely introduces the currently landscape and severity of DFI. However, it would be nice to also briefly summarize the current treatment options and their limitations in this section to further necessitate the need of discussing new clinical management approaches.

2.     Section 3 on classification of diabetic foot ulcers does not seem necessary in this review focused on DFI treatment options. A direct transition from section 2 to section 4 seems natural. The authors can consider removing this section from the review.

Specific Comments

None

Author Response

We thank the reviewer for careful review and positive responses to our review submission. We have revised the submission as follows in response: 

Comment 1

The introduction nicely introduces the currently landscape and severity of DFI. However, it would be nice to also briefly summarize the current treatment options and their limitations in this section to further necessitate the need of discussing new clinical management approaches.

Response

We thank the reviewers for positive feedback.  While novel topical treatments are the main focus of this review, we agree that some more general details in terms of the current treatment approaches helps to provide context. We have added a brief section outlining management approaches and drawbacks. Lines 146-160. 

Comment 2

Section 3 on classification of diabetic foot ulcers does not seem necessary in this review focused on DFI treatment options. A direct transition from section 2 to section 4 seems natural. The authors can consider removing this section from the review.

Response.  

The section outlining classification of infections is important in the context of patient management and treatment because infection severity, although complex to classify,  can help to predict poor clinical outcomes.  However, we agree that this section could be better explained and simplified to avoid over-emphasis and distraction from the main goals.  We have revised this section and removed the table (also in response to Reviewer 2 comment).

Reviewer 2 Report

Turzanska et al have reviewed the current treatments being used for the management of diabetic foot infections and have suggested some potential treatments that show promise. The authors have focussed on a few types of treatments for example antimicrobial peptides. The authors need to emphasise why they have decided to focus on these treatments specifically. Overall the review provides a well-written up to date summary of  different treatments with their advantages and limitations. Minor adjustments need to be made prior to publication as listed below: 

Table 1- needs a legend underneath explaining the contents e.g. what does ‘grade’ mean and what do P1, E, etc actually mean. Lines 83-86 need to be explained in more detail here for the reader for clarification. The table itself could be arranged better e.g. what symptoms do the grades refer to e.g. it is not clear what symptoms/ lines of text grade I2 involves. Perhaps put extra lines in to separate the different sections?

Figure 1 -title needs to be more descriptive e.g. mention the significance of these SNAPPS e.g. as a therapeutic. Figure is messy, needs to be bigger and have the four boxes align with one another to maximise the use of space. For example a can be bigger and d can be wider so lines up with figures a, b and c.

Table 2 needs a proper legend underneath explaining what is being shown in the table.

Consistency needed in use of capital letters e.g. names of chemicals in table 2.

Line 84 – typo should say modification (was a modified of the PEDIS classification system)

Line 120 – However- no capital

Author Response

We thank the reviewer for a careful review and positive responses to our review submission. We have revised the submission in response to theses comments as follows:

Comment 1

Figure 1 -title needs to be more descriptive e.g. mention the significance of these SNAPPS e.g. as a therapeutic. Figure is messy, needs to be bigger and have the four boxes align with one another to maximise the use of space. For example a can be bigger and d can be wider so lines up with figures a, b and c.

Response – The figure has been amended and the title modified to be more descriptive

Comment 2

Table 1- needs a legend underneath explaining the contents e.g. what does ‘grade’ mean and what do P1, E, etc actually mean. Lines 83-86 need to be explained in more detail here for the reader for clarification. The table itself could be arranged better e.g. what symptoms do the grades refer to e.g. it is not clear what symptoms/ lines of text grade I2 involves. Perhaps put extra lines in to separate the different sections?

Response

Lines 83-86 have been rewritten for clarification

Reviewer 1 felt that the section on wound classification could be omitted.  We have responded to this, with a better explanation of why wound classification systems are important (but complex) in patient management, as wound and infection severity are important parameters that may predict poor outcomes.  However, to focus more on the novel antimicrobial aspects, we have removed the table as it is one that is openly available.  We have retained the references to the table for the reader.

Comment 3

Table 2 needs a proper legend underneath explaining what is being shown in the table.

 Consistency needed in use of capital letters e.g. names of chemicals in table 2.

Response

Footnotes have been added to columns 2,3 and 4 to summaries what is shown.  Names have been checked for consistent use of capitals.